# Uncertainty Preservation in Generative Visual Autoregression

## Abstract

Autoregressive (AR) models are among the most popular frameworks for visual generation. Recently, they have demonstrated competitive performance in visual generation through next-scale prediction. However, the training pipeline relies on cross-entropy as an objective, which enforces a precise target distribution for each token. This can lead to overconfident, sharp predictions and ultimately, a lack of diversity or mode collapse in the generated samples. In supervised learning, techniques such as label relaxation have been proposed to address the shortcomings of cross-entropy by replacing precise targets with sets of plausible distributions. However, in generative models, uncertainty should be considered as dispersion over a range of plausible outcomes rather than ambiguity about a single correct label. Building on this view, we introduce *uncertainty preservation* for visual autoregressive models. Specifically, we obtain predictive distributions as *second-order likelihoods* and penalize any deviations in their dispersion from a calibrated reference. Moreover, we implement a semantic entropy loss to provide a complementary measure of uncertainty that aligns with consistency at the meaning level. In theory, we demonstrate that our approach is a special case of second-order regularization, whereby penalizing variance deviation is equivalent to controlling the scale component of a Wasserstein distance between credal distributions. Through extensive experiments on multiple settings and datasets, we demonstrate that our model, despite its simplicity and low computational overhead, improves generation quality and diversity.

## 1 Introduction

Autoregressive models have emerged as strong contenders in generative modeling. Recently, Visual Autoregressive (VAR) models (Tian et al., 2024; Guo et al., 2025) have demonstrated competitive performance compared to diffusion-based generative approaches in terms of both computational efficiency and fidelity. However, the training of VAR models still depends on cross-entropy, which enforces a precise distribution for each token prediction. This can result in excessively sharp and overconfident posteriors that fail to capture the natural dispersion present in the data adequately. Consequently, this often leads to reduced sample diversity, mode collapse, and under-utilization of the shared codebook. These limitations highlight the need for training objectives that extend beyond first-order likelihoods and explicitly *preserve uncertainty* at various levels of the generation process.

In predictive supervised learning, methods such as label smoothing (Müller et al., 2019) and label relaxation (Lienen & Hüllermeier, 2021b) have been proposed to mitigate the overconfidence induced by cross-entropy by replacing point targets with softened or credal sets of distributions. These methods estimate uncertainty with respect to a known ground-truth label. In contrast, generative modeling does not rely on label-level ground truth: the objective is rather to match the distribution of observed outcomes, such as natural images in vision or human-produced text in language. Therefore, uncertainty in generative tasks should be viewed not as ambiguity about a single correct label, but as dispersion over a range of plausible outcomes. This perspective motivates the use of *second-order distributions*, which can represent and *preserve uncertainty at the distributional level*, rather than collapsing it into overly sharp token-level likelihoods.

In parallel, recent studies in natural language generation have highlighted that first-order likelihoods are insufficient to capture meaningful uncertainty. For instance, *semantic entropy* (Kuhn et al.,

2023) was introduced to quantify uncertainty at the level of meaning rather than surface forms. By clustering paraphrases that share identical semantics, semantic entropy preserves uncertainty across equivalence classes of generations, providing a more faithful measure of model confidence. This perspective suggests an important analogy for visual generation: rather than distributing probability mass arbitrarily across redundant codes, autoregressive vision models should aim to preserve uncertainty in a way that reflects semantic similarity in the codebook and supports coherent diversity in the generated samples.

To this end, we propose a principled framework for *uncertainty preservation* in visual autoregression. Our method enhances standard likelihood training by incorporating second-order regularization, motivated by second-order distributions (Barrett & Lampard, 1955; Walley, 1997; Sale et al., 2024) and imprecise probability theory (Walley, 1997; Augustin et al., 2014), to explicitly constrain the dispersion of predictive distributions relative to calibrated references. This mitigates the collapse of token-level variability and preserves diversity across scales. Additionally, we adapt the concept of *semantic entropy* (Kuhn et al., 2023) to the vision domain, ensuring that predictive uncertainty aligns with meaningful equivalence classes in the codebook rather than being arbitrarily spread across redundant codes. Together, these components encourage VAR models to strike a balance between fidelity, robustness, and diversity, providing both theoretical grounding through Wasserstein-based measures of second-order uncertainty and practical improvements in large-scale image generation.

Empirically, we evaluated our approach on AFHQ (Choi et al., 2020) and ImageNet-100 (Deng et al., 2009a), demonstrating consistent quantitative and qualitative improvements. The experimental results show that our proposed regularized losses reduce the FID from 60 to 46, indicating a significant improvement in image generation quality. We evaluate our model on various downstream tasks, including inpainting, outpainting, and class-conditional editing. Our method achieves a superior FID score for image generation and calibration, outperforming the baseline in various experiments on large datasets. This confirms our model's scalability and practical applicability in real-world scenarios.

To summarize, our contributions are as follows: (i) We propose a **second-order objective** for Visual autoregression, that preserves predictive dispersion by penalizing deviations from calibrated reference distributions in the codebook. (ii) We introduce **semantic entropy** from language modeling to visual generation, implementing consistency across semantically equivalent token reconstructions and mitigating codebook under-utilization. (iii) Theoretically, we explain the connection between variance penalization and Wasserstein geometry, indicating that our loss corresponds to controlling the scale component of Wasserstein deviations under the codebook-induced ground metric. (iv) Empirically, we show that our simple and efficient method for preserving uncertainty achieves competitive performance quantitatively.

## 2 METHOD

### 2.1 PRELIMINARY

The visual autoregressive modeling (Tian et al., 2024) adapts conventional autoregressive models to a next-scale prediction framework, where each autoregressive step predicts an entire token map instead of a single token. Given an input image $x \in \mathbb{R}^{H \times W \times C}$, and a multi-scale quantized autoencoder $E(\cdot)$, VAR encodes it into a hierarchy of discrete latent token maps $R = \{r_1, r_2, \ldots, r_K\}$, where each $r_k \in [V]^{h_k \times w_k}$ is the token map at resolution $(h_k, w_k)$ with vocabulary size $V$. VAR learns the joint distribution over all scales using the next-scale factorization:

$$p(R) = p(r_1, r_2, \ldots, r_K) = \prod_{k=1}^{K} p(r_k \mid r_{<k}), \tag{1}$$

where $r_{<k} = \{r_1, \ldots, r_{k-1}\}$.

To obtain the token hierarchy $R$, VAR employs a multi-scale VQ-VAE (Van Den Oord et al., 2017). The encoder $E(\cdot)$ maps the image to a latent representation $f \in \mathbb{R}^{h \times w \times C}$, which is progressively quantized into discrete token maps $\{r_k\}$ at multiple resolutions. A shared codebook $Z \in \mathbb{R}^{V \times C}$ contains the embedding vectors, and a decoder $D(\cdot)$ reconstructs the image from the quantized embeddings. Sharing a codebook across all scales ensures that each token map $\{r_k\}$ draws from the same vocabulary, which is essential for achieving coherent multi-scale autoregression.

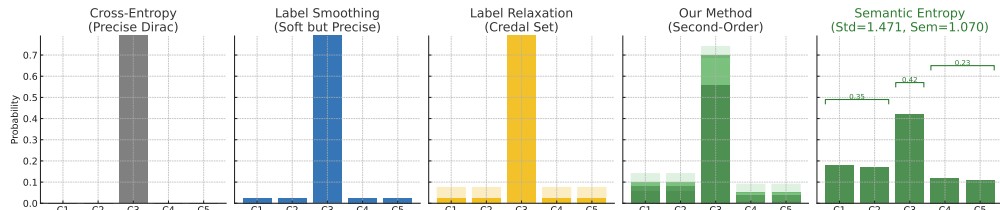

Figure 1: Comparison of different training objectives; Cross-Entropy (shown in gray) enforces a precise target, leading to overconfident predictions. Label Smoothing (blue) (Szegedy et al., 2016) softens the target while maintaining a single, precise distribution. Label Relaxation (yellow) (Lienen & Hüllermeier, 2021b) replaces the precise distribution with a credal set, allowing bounded uncertainty. Our method (shown in green) introduces second-order uncertainty, modeling dispersion around predictive distributions, and further incorporates semantic entropy to preserve consistency across semantically equivalent generations.

**Training objective** The VAR model employs the cross-entropy loss between the predicted categorical distribution and the ground-truth token maps as its training objective:

$$\mathcal{L}_{\text{CE}} = -\sum_{k=1}^{K} \sum_{i=1}^{h_k w_k} \log p_\theta \big( r_k^{(i)} \mid r_{<k} \big), \tag{2}$$

where $r_k^{(i)}$ represents the token at position $i$ in the scale $r_k$, and $p_\theta(\cdot \mid r_{<k})$ is the predicted distribution at that position. While $\theta$ denotes the parameters of the autoregressive model (i.e., the learnable weights of the neural network).

**Limitations** The standard cross-entropy objective implicitly assumes that the predictive distribution $p_\theta(r_k \mid r_{<k})$ is a precise representation of the ground truth. This enforcement of 'first-order' fidelity can lead to overconfident predictions, hindering robustness and potentially causing generative mode collapse (Shabgahi et al., 2025; Gerstgrasser et al., 2024; Srivastava et al., 2017) (Please refer to the first plot in Figure 1). In practice, it ignores second-order properties such as the variability or confidence in the probability estimates. As a result, VAR models tend to produce overconfident distributions, which hinders robustness in generative modeling and may lead to *generative mode collapse*. Moreover, sharing a VQ-codebook across different scales causes gradients to select a small group of codes repeatedly. This leaves many other codes *unused*, which reduces the diversity of the codebook. This lack of diversity can lead to *codebook collapse* (Zhao et al., 2024; Baykal et al., 2024), where the model struggles to learn and represent diverse features.

To overcome these limitations, we propose an *uncertainty-preserving* approach in which predictive distributions are modeled as second-order likelihoods rather than precise estimates. We regularize their dispersion relative to calibrated references, ensuring that confidence levels stay consistent with the generative task. Moreover, we incorporate semantic entropy regularization to align uncertainty with meaningful codebook structures, thereby mitigating both mode and codebook collapse. Together, these components encourage VAR models to enhance diversity while improving robustness and fidelity in visual generation without additional computational overhead.

## 2.2 MOTIVATION

The problem of overconfident predictions under cross-entropy training is not limited to generative modeling; it has also been recognized in discriminative learning. For example, in a classification task, cross-entropy encourages models to produce overly sharp posteriors, which is mitigated via *label smoothing* (Szegedy et al., 2016; Müller et al., 2019). While label smoothing has proven more effective than relying on hard labels, it still relies on one single distribution, making it a precise but potentially inaccurate replacement for the target distribution. More principled approaches such as *label relaxation* (Lienen & Hüllermeier, 2021b) argue for replacing precise targets with *credal sets* of distributions, aligning naturally with the framework of *imprecise probabilities* (Walley, 1991; Augustin et al., 2014) that has received considerable attention both in the theoretical machine learning (Chau et al., 2025; Singh et al., 2024; Caprio et al., 2024; 2023b; Rodemann & Augustin, 2022; 2021; Dietrich et al., 2024; Jansen et al., 2024; 2023b;a; Sale et al., 2024; 2023; Gong & Meng, 2021)

and the applied deep learning (Löhr et al., 2025; Dutta et al., 2025; Garces-Arias et al., 2025; Wang et al., 2024; Caprio et al., 2023a; Marquardt et al., 2023; Lienen & Hüllermeier, 2021a) community recently. This perspective acknowledges that there is not always a single correct target distribution, but rather a range of plausible ones.

Our work builds on this perspective in the context of generative visual autoregression. Here, the notion of *uncertainty preservation* becomes central: instead of estimating an unknown ground truth label, the objective is to maintain the appropriate level of dispersion across predictive distributions. Such dispersion across different distributions is a natural motivation for *second-order distributions*, i.e., distributions over distributions.

As touched upon above, prior work on *label relaxation* Lienen & Hüllermeier (2021b) has employed *credal sets*, i.e., sets of probability measures and corresponding first-order distributions, to model second-order uncertainty in predictive tasks. For generative ones like VAR we move beyond credal sets and consider explicit information about the first-order distributions' probabilities, as these latter encapsulate the appropriate level of dispersion across predictive distributions—pivotal to uncertainty preservation. Since credal sets represent multiple first-order distributions as unstructured sets, they lack any information about these distributions' weights. This is why we move beyond credal sets and explicitly incorporate second-order distributions into VAR. This translates to extending likelihood-based training (as detailed below) and soften its rigid commitment to precise predictions. The conceptual link to label relaxation provides a principled motivation for our proposed second-order method, which aims to address overconfidence, mode collapse, and codebook underutilization in a unified manner.

## 2.3 Uncertainty Preservation

These limitations suggest that precise probability objectives such as cross-entropy (and even credal sets of those) are insufficient for robust visual autoregression. We propose a simple and efficient way to incorporate **uncertainty preservation** into the training pipeline of visual autoregressive models. By introducing second-order distributions that explicitly model the dispersion of predictive distributions. Our goal is to maintain the level of predictive dispersion that is appropriate for each scale and consistent with the data manifold. This requires moving beyond first-order likelihoods to a second-order formulation that constrains not only the expected probabilities but also their variability.

**Second-order formulation** At each position $(i, k)$, the model outputs a categorical distribution $p_\theta^{(i)} \in \Delta^{V-1}$. Standard cross-entropy constrains only the likelihood assigned to the ground-truth token, leaving the overall sharpness of $p_\theta^{(i,k)}$ uncontrolled. To maintain an appropriate level of dispersion, we present a second-order description of predictive behavior by considering functionals of $p_\theta^{(i)}$ beyond its mean assignment. Specifically, we monitor the dispersion defined as:

$$\sigma_\theta^2(i, k) = \text{Var}_{v \in [V]}\big[p_\theta^{(i)}(v)\big], \tag{3}$$

which reflects how concentrated or diffuse the distribution is across the vocabulary. However, penalizing variance in isolation would simply encourage collapse; instead, a reference distribution $q^*$ is required, which induces the target variance $\sigma_\star^2(k)$ and serves as the calibration anchor against which the model's predictions are compared.

**Reference calibration** To calibrate second-order uncertainty, we introduce a *reference distribution* $q^*$, which encodes the expected level of dispersion at each scale $k$. For example, $q^*$ may correspond to: (i) a maximum-entropy distribution (uniform probabilities) that enforces maximal diversity across tokens; (ii) a dataset-calibrated distribution that matches average entropy at scale $k$ that anchors dispersion to the empirical uncertainty observed in real data, or (iii) a semantic-aware reference distributions leverage the geometry of the codebook embeddings, encouraging the model to allocate probability mass not just to the predicted code, but also to nearby, semantically similar codes. This serves as a calibration anchor, ensuring that predictive dispersion (uncertainty) is preserved rather than arbitrarily collapsed. We opt for an approximately normal distributions with fixed location parameter and minimal variance as $q^*$ later.

**Second-order regularizer** We penalize the variance relative to a fixed *reference variance* $\sigma_\star^2(k)$, which specifies the preferred level of dispersion. Assuming the same location parameter (e.g.,

mean) and elliptical distributions (e.g., normal distribution), this penalization then corresponds to the Wasserstein distance to the reference distributions (with reference variance). We define discrepancy between predicted and reference variances as:

$$\mathcal{L}_{\text{2nd}} = \sum_{k=1}^{K} \sum_{i=1}^{h_k w_k} d\big(\sigma_\theta^2(i,k),\ \sigma_\star^2(k)\big),$$
(4)

where $\sigma_\star^2(k)$ indicates the variance at scale $k$ and $d(\cdot,\cdot)$ is a discrepancy measure. In practice, $d$ can be instantiated as the squared difference, the KL divergence, or a Wasserstein distance, the latter providing a geometry-aware interpretation of uncertainty preservation.

*Remark.* It is important to emphasize that minimizing variance alone would undesirably drive $\sigma_\theta^2(i,k)$ toward zero, resulting in collapsed and overconfident distributions. The connection to Wasserstein geometry occurs only when variance is penalized relative to a fixed *reference variance* $\sigma_\star^2(k)$, which specifies the preferred level of dispersion, assuming the same location parameter (e.g., mean) and elliptical distributions (e.g., normal distribution). In this case, Eq. 4 can be interpreted as controlling the scale component of the Wasserstein distance between the predicted second-order distribution $Q$ and the calibrated reference distribution $Q^\star$. Using the approximate normal distribution with a fixed location parameter and minimal variance, as mentioned above, as a reference $q^*$, will allow us to efficiently approximate $\mathcal{L}_{\text{2nd}}$ (see proposition 3.1 and proposition 3.2).

**Semantic entropy regularization**    Uncertainty preservation must also respect semantic equivalence. To ensure that uncertainty aligns with semantics rather than token redundancy, we adapt *semantic entropy* from natural language generation to the visual domain. Let $\mathcal{C}$ denote equivalence classes of code sequences that reconstruct the same semantic content. The semantic entropy is then defined as:

$$H_{\text{sem}}(p_\theta) = -\sum_{c \in \mathcal{C}} \left(\sum_{s \in c} p_\theta(s)\right) \log \left(\sum_{s \in c} p_\theta(s)\right),$$
(5)

which aggregates probabilities of semantically equivalent sequences before computing entropy. Minimizing this entropy encourages the model to distribute probability mass consistently across reconstructions that share the same meaning.

**Our objective**    The total objective combines first-order accuracy, second-order calibration, and semantic consistency as:

$$\mathcal{L} = \mathcal{L}_{\text{CE}} + \lambda_1\, \mathcal{L}_{\text{2nd}} + \lambda_2\, \mathcal{L}_{\text{SE}},$$
(6)

where $\mathcal{L}_{\text{CE}}$ is the standard cross-entropy loss, $\mathcal{L}_{\text{2nd}}$ enforces variance calibration, $\mathcal{L}_{\text{SE}}$ corresponds to semantic entropy regularization, and $\lambda_1, \lambda_2$ are hyperparameters that balance the contributions of these different terms.

## 3 THEORETICAL JUSTIFICATION

In this section, we provide theoretical justification for our proposed uncertainty-preserving method. We formalize predictive variability using second-order distributions, connect variance penalization to Wasserstein geometry on the codebook, and show how this yields a principled surrogate for controlling dispersion in autoregressive generation. These results ground our loss design in well-defined probabilistic and geometric principles.

### 3.1 SECOND-ORDER UNCERTAINTY

Consider the set of possible codebook $Y = \{1, \ldots, V\}$. For a given image $x$, the VAR factorization provides $p_\theta(r_k \mid r_{<k})$ over token maps. At a specific location $(i,k)$ we denote the **first-order** categorical vector $\theta^{(i,k)} \in \Delta^{V-1}$. To define both aleatoric and epistemic uncertainties, we elevate this to a **second-order** object $Q^{(i,k)} \in \mathcal{P}(\Delta^{V-1})$, a distribution over first-order distributions.

### 3.2 GEOMETRY ON THE CODEBOOK

Considering $e_v \in \mathbb{R}^d$ as the embedding of code $v$. We define a ground metric on $Y$:

$$d_0(u,v) := \|e_u - e_v\|_2 \quad \text{(or cosine distance)}$$
(7)

This induces a 1-Wasserstein metric $W_1^{(d_0)}$ on $\Delta^{V-1}$ (the space of first-order categorical distributions), and, by lifting once more, a **second-order** Wasserstein metric $\mathbb{W}_1^{(d_0)}$ on $\mathcal{P}(\Delta^{V-1})$. This geometry respects semantic proximity between codes and provides a codebook-aware analogue of total uncertainty, aleatoric uncertainty, as well as epistemic uncertainty distances.

*Remark.* If $d_0$ is the trivial $0-1$ cost, then $W_1^{(d_0)} = \frac{1}{2}\|\cdot\|_1$, recovering closed forms used in label-space formulations.

### 3.3 SECOND-ORDER CALIBRATION

We instantiate $Q$ with a Dirichlet family $Q = \mathrm{Dir}(\alpha)$. This yields closed-form first-order moments $\bar{\theta} = \alpha/\alpha_0$ and tractable expectations such as $\mathbb{E}_Q[H(\theta)]$.

*Proposition 3.1* (Wasserstein ≈ Mean/Scale Matching). If we approximate first-order $\theta$ by Gaussian in logit space and $Q, Q^\star$ by Gaussian families, then

$$\mathbb{W}_2^2(Q, Q^\star) = \|\mu - \mu^\star\|^2 + \mathrm{Tr}\left(\Sigma + \Sigma^\star - 2(\Sigma^{1/2}\Sigma^\star\Sigma^{1/2})^{1/2}\right).$$

Therefore, minimizing $\mathbb{W}_2$ decomposes into mean and covariance matching. Variance penalization is consequently equivalent to controlling Wasserstein deviation.

*Proposition 3.2* (Variance–Wasserstein connection). Under a Gaussian approximation of predictive distributions, the squared 2-Wasserstein distance between a predicted second-order distribution $Q$ and a reference $Q^*$ decomposes into a mean-matching term and a variance-matching term. The variance-based loss in Eq. 4 is therefore a tractable surrogate for the variance component of this Wasserstein distance.

## 4 EXPERIMENTS AND RESULTS

### 4.1 CONFIGURATION AND SETUPS

**Datasets** The datasets utilized in our experiments are as follows: The AFHQ dataset (Choi et al., 2020) comprises 15,000 high-resolution images of size 512×512 pixels, covering three categories: cats, dogs, and wild animals. Each class contains approximately 5,000 images. This dataset is popular for evaluating generative models due to its diversity and fine-grained visual details. The **ImageNet-100** (Deng et al., 2009b), which is a smaller subset of the ImageNet but with 100 classes, maintains the exact image resolution of 256 x 256. It provides a more computationally feasible benchmark while still preserving substantial diversity across object categories. The Databricks-dolly-15k Conover et al. (2023) dataset, which contains public information (e.g., some information from Wikipedia), and we used this dataset to examine the impact of our method for the NLP task.

**Baseline** We use the standard Visual Autoregressive (VAR) model (Tian et al., 2024) with the cross-entropy objective as a baseline. Moreover, we use Qwen 1.5 Yang et al. (2025) as a baseline for downstream task evaluation, where the model is trained with a standard cross-entropy loss for language modeling.

**Metrics** We assess generation quality and diversity using standard criteria: **Fréchet Inception Distance (FID ↓)** to measure distributional alignment with real images, **Inception Score (IS ↑)** to evaluate both fidelity and diversity. Moreover, we report the prediction performance on robustness tasks with the following metrics: **Accuracy ↑**: refers to the proportion of test observations that are precisely predicted by the model's outcome as belonging to the correct class. **Negative log-likelihood (NLL) ↓**: measures the probability of observing the given test data given the estimated model parameters, multiplied by -1. This measure quantifies the degree to which the model's estimated parameters fit the test observations. **Expected calibration error (ECE) ↓** (Naeini et al., 2015): calculated as the mean absolute difference between the accuracy and confidence of the model's predictions, where confidence is represented as the highest posterior probability among the predicted classes. The difference is calculated across equally spaced confidence intervals or bins and is weighted by the relative number of samples in each bin. **AUROC ↑**: the area under the ROC curve describes the relationship between false-positive and false-negative rates for various classification thresholds. In this case, the positive and negative classes refer to whether an observation is in or out of a given distribution, respectively, and the ROC curve is plotted as the threshold for classifying an observation as "positive" is gradually increased.

**Tasks** We evaluate our model using a wide range of conditional generation tasks: in-painting (predicting and synthesizing missing regions with semantically consistent content), out-painting (extending images beyond their original boundaries), and class-conditional editing, where images are modified according to target class labels. These tasks follow the experimental protocol of the original VAR paper (Tian et al., 2024), ensuring comparability with established baselines.

**Training** We follow VAR's (Tian et al., 2024) training setup, training our models at an image resolution of $256 \times 256$ pixels in the latent space of their pre-trained VQVAE (Van Den Oord et al., 2017; Esser et al., 2021) (with codebook size $V = 2048$, and resolution stages $1^2, 2^2, 3^2, 4^2, 5^2, 6^2, 8^2, 10^2, 13^2, 16^2$). We train *VAR-d16* models with a transformer depth of 16 layers unless noted otherwise. Our models are trained for 400k steps using AdamW with decay $= 0.05$ with $(\beta_1, \beta_2) = (0.9, 0.95)$ at a batch size of 16 for AFHQ and 768 for ImageNet100 with a learning rate of $1e - 4$.

**Sampling** Unless noted otherwise, we follow standard sampling settings (Tian et al., 2024) and use top-k (Fan et al., 2018) sampling with $k = 900$, top-p sampling (Holtzman et al., 2019) with $p = 0.95$, and a classifier-free guidance (CFG; Ho & Salimans) scale of 1.5 for evaluations on ImageNet100 and 8.0 for AFHQ. For qualitative samples, we use a classifier-free guidance scale of 3.0 following standard practice.

## 4.2 RESULTS

We follow the experimental protocol of prior VAR models, given our computational resources. Our analysis covers quantitative metrics, scaling and efficiency trends, and downstream conditional generation tasks, including in-painting and out-painting. Across all settings, uncertainty preservation consistently improves generation fidelity, robustness, and diversity compared to the cross-entropy baseline, as we demonstrate in our qualitative results.

Table **??** summarizes results on ImageNet-100 and AFHQ. Our method achieves consistent improvements in FID and IS on two image datasets. Importantly, on AFHQ, it reduces FID from 58 to 46 compared to the VAR baseline while improving IS. These results demonstrate that second-order calibration and semantic entropy regularization yield tangible improvements in both perceptual quality and semantic consistency, without increasing inference cost.

*Image out-painting* Out-painting experiments are shown in Figure 2. The baseline (columns 5,6, and 7) often produces oversimplified extrapolations or repetitive patterns due to collapsed uncertainty, whereas our method (columns 2,3, and 4) generates diverse continuations that remain consistent with the input context. These findings provide empirical evidence for our theoretical claim that uncertainty preservation enables robust extrapolation to novel contexts, thereby enhancing the creative capacity of autoregressive visual models.

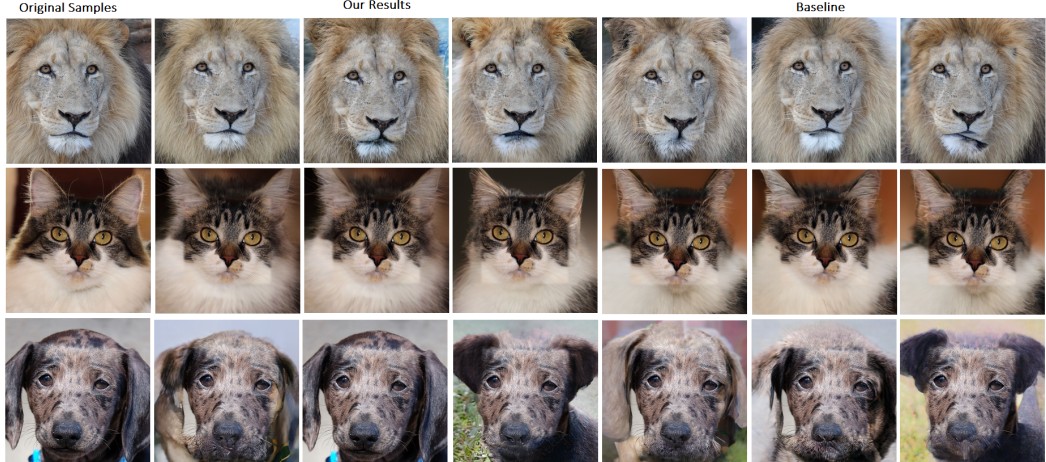

Figure 2: Out-painting evaluation in downstream tasks, the images in the first column are taken from the AFHQ testing set, while the images in columns 2, 3, and 4 are generated by our methods. The samples in columns 5, 6, and 7 are generated by the baseline.

*Image in-painting* In-painting results are reported in Figure 3. Our model generates completions that are sharper and exhibit smoother transitions at mask boundaries compared to the cross-entropy baseline. Importantly, semantic entropy regularization prevents collapsed or repetitive completions, yielding diverse and semantically plausible fills. These improvements hold across both structured (ImageNet) and unstructured (AFHQ) domains.

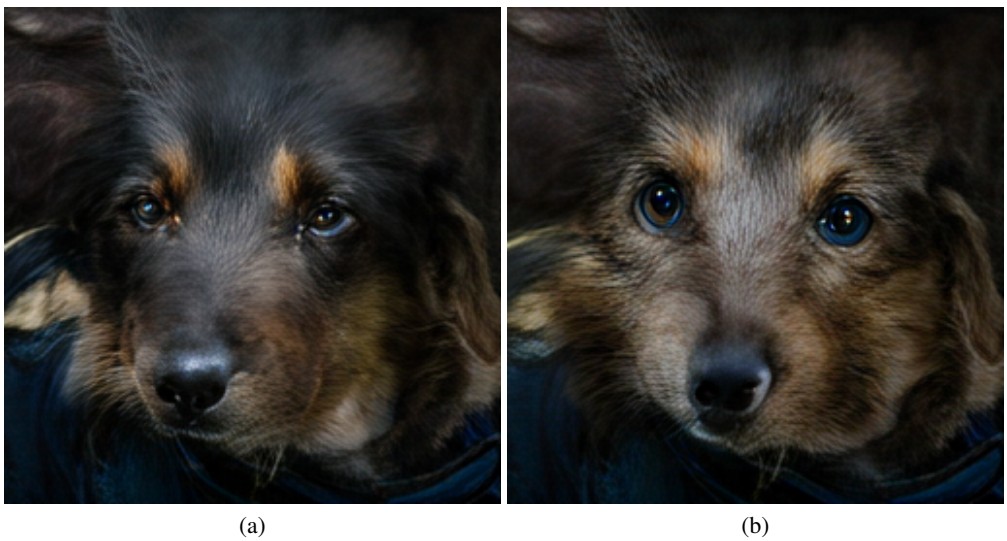

(a)  (b)

Figure 3: Comparison of generative editing techniques on a dog image. (a) Inpainting result, with missing regions filled based on the surrounding context. (b) Outpainting result, where the original image is extended beyond its boundaries to synthesize new plausible content.

Inpainting and outpainting are two distinct image synthesis techniques. Inpainting aims to reconstruct or fill missing regions in an image using contextual information from surrounding pixels. Outpainting, on the other hand, extends the original image beyond its existing boundaries, generating additional plausible content consistent with the scene. As illustrated in Figure 3 (a) and (b), inpainting preserves the integrity of localized features, while outpainting produces a more expansive representation by synthesizing new visual information.

*Scaling and efficiency analysis* We further investigate how our method behaves under model scaling. Following prior VAR scaling law studies, we train models with depths ranging from 16 to 20 layers. Our second-order objective shifts the scaling exponent, indicating improved sample efficiency. In other words, comparable test losses can be achieved with smaller models, suggesting that uncertainty preservation enhances the effective utilization of model capacity. The results demonstrated in Table 2 indicate that the impact of our uncertainty preservation on the autoregressive architecture yields consistent gains across embedding dimensions $d = 16$ and $d - 20$. For both AFHQ and ImageNet-100, the proposed method achieves lower FID and higher IS than the corresponding VAR baselines, indicating improved sample fidelity and diversity. Notably, the improvements become more pronounced at higher dimensionality, where the model with $d = 20$ attains the best overall performance, suggesting that uncertainty-aware representations scale favorably with model capacity.

Table 2: Comparison of our uncertainty-preserving autoregressive model with the VAR baseline across different model depths. We report Fréchet Inception Distance (FID, ↓) and Inception Score (IS, ↑). Our method consistently improves over the baseline at both depths, with the best results highlighted in **bold**.

| Model | Method | AFHQ (FID) | AFHQ (IS) | ImageNet-100 (FID) | ImageNet-100 (IS) |
|-------|--------|------------|-----------|--------------------|--------------------|
| d16 | Baseline | 54.59 | $4.18 \pm 0.19$ | 20.45 | $15.20 \pm 0.82$ |
|  | Our Method | **46.97** | **$4.43 \pm 0.16$** | **20.02** | **$15.86 \pm 0.54$** |
| d20 | Baseline | 46.12 | $4.39 \pm 0.17$ | 17.28 | $16.42 \pm 0.71$ |
|  | Our Method | **39.68** | **$4.72 \pm 0.14$** | **16.91** | **$17.15 \pm 0.48$** |

*OOD robustness* is the ability of a model to recognize test samples from classes that were not present during training is evaluated using OOD detection, as discussed in Geng et al. (2020). We conduct experiments on ImageNet-O (Srivastava et al., 2022) to assess the model's generalization from IND to OOD datasets, as well as to estimate the model's uncertainty on OOD datasets. Note that evaluation is performed directly after pretraining without a fine-tuning step. Table 3 shows and compares the results for the OOD task in terms of AUROC.

Table 3: Out of distribution detection and corrupted dataset: AUROC for out of distribution detection for Imagent-O dataset, where higher AUROC is better and lower NLL, and mCE for Imagent-C dataset, where lower mCE is better.

| Method | NLL ($\downarrow$) | AUROC (%) ($\uparrow$) | mCE ($\downarrow$) |
|---|---|---|---|
| VAR | 74.6 | 69.7 | 68.0 |
| Our Method | 72.0 | 73.5 | 71.7 |

*Corrupted task analysis* is an essential aspect of model robustness is its capability to produce precise predictions when the test data distribution changes. We examine model robustness in the context of *covariate shift*. Table 3 presents the improved performance metrics. Our method outperforms the baseline and has comparable predictive performance to the baseline for mCE.

## 4.3 ABLATION ANALYSIS

To gain a deeper understanding of the behavior and performance of our proposed method, we conducted several ablation analyses to study various aspects of our approach. Specifically, we investigate the following factors: (i) the role of the loss component in training; (ii) computational efficiency; (iii) the impact of our proposed loss in model calibration (meanwhile of downstream tasks) and for natural language processing (NLP). and (iv) evaluation of sample diversity C.1, which we present together with qualitative analysis in Appendix C.2

**Impact of different components of uncertainty preservation loss** We conduct a series of ablation analyses to study the role of each loss component. Based on Fig 4, our analysis indicates that both $\lambda_1$ (second-order regularization) and $\lambda_2$ (semantic entropy) play an essential role in enhancing performance when properly tuned. In particular, moderate values of $\lambda_1$ (e.g., $0.001 \leq \lambda_1 \leq 0.003$) achieve the lowest FID, confirming the benefit of variance calibration without over-regularization. Similarly, semantic entropy regularization ($\lambda_2$) consistently reduces FID and enhances guidance consistency, while `cfg` values remain stable across the best-performing settings, highlighting robustness in controllability.

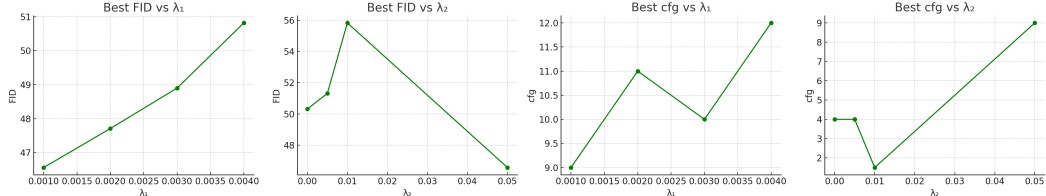

Figure 4: Study of hyperparameters of our objective function on AFHQ dataset.

**Computational Cost and Efficiency Analysis** Table 5 shows the computational efficiency of our approach on the ImageNet-100 dataset. As we mentioned earlier, our method incurs zero additional overhead in terms of training time or GPU memory usage compared to the baseline, despite integrating second-order and semantic entropy regularization. This validates the practicality of our framework, demonstrating that significant performance gains can be achieved without requiring extra computational resources or architectural modifications.

**Impact of our loss functions on downstream tasks and NLP datasets** In this ablation analysis, we further assess the influence of the proposed loss functions on natural-language generation quality and downstream task performance. The results in Table 6 show that integrating our loss formulation substantially enhances distributional alignment and robustness in generation: the MAUVE score in-

Table 5: Computational cost in $8 \times$ DGX-A100 80G GPUs (PyTorch) on ImageNet-100.

| Method | Parameters(M) | Batch-size | Memory / GPU | Time / 200-ep. |
|---|---|---|---|---|
| Baseline | 300 | 768 | 74 G | 27 (h) |
| Our method | 300 | 768 | 74 G | 27 (h) |

creases markedly compared to zero-shot and SFT baselines, indicating that the learned text distribution more closely matches human data. Simultaneously, the reduced Self-BLEU highlights improved sample diversity, mitigating mode collapse. These gains translate into stronger representations for downstream NLP tasks, as the loss functions encourage both semantic fidelity and variability during training. Consequently, models fine-tuned with our objective not only generate higher-quality text but also exhibit improved generalization across standard NLP benchmarks, demonstrating the broader utility of the proposed losses beyond pure generation metrics.

Table 6: Text generation quality evaluation. MAUVE Score measures distributional similarity to human text (higher is better), while Self-BLEU measures generation diversity (lower is better).

| Method | MAUVE Score $\uparrow$ | Self-BLEU $\downarrow$ |
|---|---|---|
| Zero-shot | 0.5127 | 0.0858 |
| SFT Finetuned | 0.7744 | 0.0651 |
| **Our Method** | **0.9537** | **0.0630** |

## 5 CONCLUSION AND FUTURE WORKS

In this work, we proposed *uncertainty preservation* to address limitations of cross-entropy training in visual autoregressive models, which enforce overly precise predictions and suppress the natural dispersion of generative distributions. Our framework augments standard likelihood training by incorporating second-order regularization and semantic entropy. Our theoretical analysis demonstrated a principled connection between variance penalization and Wasserstein geometry, showing that the proposed second-order loss corresponds to controlling the scale component of a Wasserstein distance between distributions. Empirically, our method demonstrated consistent improvements on image datasets, reducing FID and enhancing diversity with low computational overhead. These results highlight the practical benefits of incorporating second-order uncertainty in autoregressive generation. Beyond image synthesis, we believe that uncertainty preservation provides a unifying principle for generative modeling, opening avenues for applications in domains where robustness and diversity are crucial. Future work includes extending the framework to other modalities, investigating alternative reference distributions, and exploring richer semantic equivalence relations for uncertainty alignment.

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

## A    BACKGROUND AND PROBLEM FORMULATION

Consider encoder of $E : \mathcal{X} \to \mathcal{R}$ which maps images into hierarchical token $R = \{r_1, \ldots, r_K\}$ and quantize over a codebook $\mathcal{Z} = \{z_v\}_{v=1}^{V}$. Convetional AR models approximate the joint distribution $p(R) = \prod_k p(r_k|r_{<k})$ by minimizing the cross-entropy (CE) loss against a ground-truth token $r_{gt}$:

$$\mathcal{L}_{\text{CE}} = \mathbb{E}_{r \sim p_{\text{data}}} \left[ - \log p_\theta(r_{gt}|r_{<k}) \right] = \mathbb{E} \left[ D_{\text{KL}}(\delta_{r_{gt}} || p_\theta) \right]. \tag{8}$$

While effective for deterministic tasks, $\mathcal{L}_{\text{CE}}$ forces the predictive distribution $p_\theta$ toward a Dirac delta $\delta_{r_{gt}}$, ignoring the aleatoric uncertainty inherent in visual generation (e.g., plausible texture variations). We identify two resulting issues:

**Variance Collapse.** By enforcing a precise target, $\mathcal{L}_{\text{CE}}$ suppresses natural dispersion, leading to $\lim_{\mathcal{L}_{\text{CE}} \to 0} \text{Var}[p_\theta(\cdot|r_{<k})] \to 0$. This results overconfident posteriors that fail to capture the semantic neighborhood of the target token.

**Codebook Collapse.** Variance collapse restricts gradient flow to single indices, leaving semantically similar codes unused. This shrinks the effective vocabulary $|\mathcal{V}_{\text{active}}| \ll V$, limiting sample diversity.

To mitigate this, we propose the objective as distributional matching subject to a dispersion constraint. We introduce a second-order regularizer to maintain variance near a calibrated reference $\sigma_\star^2$, and a semantic entropy term to ensure consistency across semantic equivalence classes $\mathcal{C}$:

$$\min_\theta \mathcal{L}_{\text{CE}} + \lambda_1 \underbrace{d(\sigma_\theta^2, \sigma_\star^2)}_{\text{Variance Constraint}} + \lambda_2 \underbrace{\mathcal{H}_{\text{sem}}(p_\theta, \mathcal{C})}_{\text{Semantic Consistency}}, \tag{9}$$

where $d(\cdot, \cdot)$ measures the deviation from the optimal dispersion required to preserve generative diversity.

## B    RELATED WORK

Previous studies have indicated that supervised training using cross-entropy can lead to overconfident predictions by enforcing a Dirac-like target distribution, which motivated techniques such as label smoothing (Szegedy et al., 2016; Müller et al., 2019). Label smoothing substitutes the one-hot target with a less confident distribution, which regularizes prediction and improves model calibration. Label relaxation (Lienen & Hüllermeier, 2021b) replaces single distributions with credal sets of plausible distributions, avoiding commitment to a single potentially wrong target and delivering a principled connection to imprecise probability theory. While these methods were developed in supervised learning with ground-truth labels, similar concerns arise in unsupervised learning as well as generative models, where the objective is to approximate data distributions rather than individual labels. For instance, Wasserstein GANs (Arjovsky et al., 2017) replace the binary cross-entropy loss with a Wasserstein distance approximation, which stabilizes training and mitigates mode collapse by encouraging the generator to fill the entire data manifold. These works in unsupervised learning and generative modeling (Yue et al., 2024; Shim, 2024) also demonstrate the importance of regularizing the cross-entropy objective to achieve robustness and diversity. In this paper, we enhance standard likelihood training by integrating second-order regularization and semantic entropy. The first technique constrains the spread of predictions against a baseline of calibrated references, while the second aligns the model's uncertainty with logical groupings within the codebook.

While the field of natural language generation already exploits uncertainty representations for training or decoding, see, e.g., (Garces-Arias et al., 2024; Ding et al., 2025), the field of computer vision mostly focuses on application of uncertainty as a post-hoc process, the task of OOD detection or stability analyses with diffusion models by perturbing inputs, time-steps, or manifolds, and scoring reconstruction agreement with the original sample.(Hendrycks & Gimpel, 2017; Liang et al., 2018; Liu et al., 2020; Lee et al., 2018; Graham et al., 2023; Chung et al., 2022; Nie et al., 2022; Xiao et al., 2023) While these methods highlight epistemic gaps, they remain focused on pixel-level similarity, which can vary significantly even when the underlying semantics are preserved (for instance, variations in contrast or brightness, or stripe inversion). As a result, image-space scores frequently fail to accurately reflect the semantic certainty that is crucial for evaluation and safety. Differently, we propose semantic entropy loss function to ensure that predictive uncertainty aligns with meaningful equivalence classes in the codebook rather than being arbitrarily spread across redundant codes.

Second-order distributions model uncertainty over *distributions* rather than samples. Classical lines of work include kernel embeddings and distribution regression (Muandet et al., 2017; Szabó et al., 2016). In meta-learning, dataset-level latents and neural processes capture task-level uncertainty (Edwards & Storkey, 2017; Garnelo et al., 2018), while Dirichlet/evidential approaches cast predictive distributions themselves as random objects (Malinin & Gales, 2018; Sensoy et al., 2018). Complementary distribution-focused perspectives formalize data–model co-adaptation as a fixed-point problem in Wasserstein space (Rodemann et al., 2024; Rodemann & Bailie, 2025) or learning as distributionally robust empirical risk minimization (Shafieezadeh Abadeh et al., 2015; Rahimian & Mehrotra, 2019; Bai et al., 2023; Lee & Raginsky, 2018).

# C    ADDITIONAL RESULTS AND ABLATION ON IMAGE DATASET

## C.1    SAMPLE DIVERSITY GENERATION

To study the effect of uncertainty preservation on generation diversity, we analyze structural and distributional metrics in Table 7. The low Mean Pairwise SSIM (0.32) and corresponding high SSIM Diversity Score (0.68) indicate significant structural variation between samples, providing strong empirical evidence that our method effectively mitigates mode collapse. The balanced Recall (0.42) and Coverage (0.45) scores demonstrate that the model successfully captures a broad support of the data distribution rather than concentrating on a few modes. Importantly, the diversity is achieved without sacrificing fidelity, as evidenced by a Density score of 0.95, which suggests the generated samples closely match the density of the real data manifold.

Table 7: Diversity Metrics Results

| Metric | Value | Good Range | Interpretation |
|---|---|---|---|
| Mean Pairwise SSIM | 0.32 | 0.2–0.4 | Lower = diverse |
| SSIM Diversity Score | 0.68 | 0.6–0.8 | Higher = diverse |
| Precision | 0.62 | 0.5–0.8 | Realistic images |
| Recall | 0.42 | 0.4–0.6 | Distribution coverage |
| F1 Score | 0.50 | 0.5–0.7 | Balance of P R |
| Density | 0.95 | 0.8–1.2 | ∼1.0 is ideal |
| Coverage | 0.45 | 0.4–0.6 | Similar to Recall |

## C.2    QUALITATIVE RESULTS ON IMAGE GENERATION AND SAMPLE DIVERSITY

Figure 5 showcases a selection of high-fidelity samples generated by our model, exhibiting sharp structural details and realistic textures across diverse semantic categories. The distinct visual characteristics observed in these samples demonstrate the model's ability to maintain meaningful diversity, effectively mitigating the mode collapse often observed in standard cross-entropy training. This qualitative evidence supports our quantitative findings, confirming that preserving second-order uncertainty enhances both perceptual quality and generative variation.

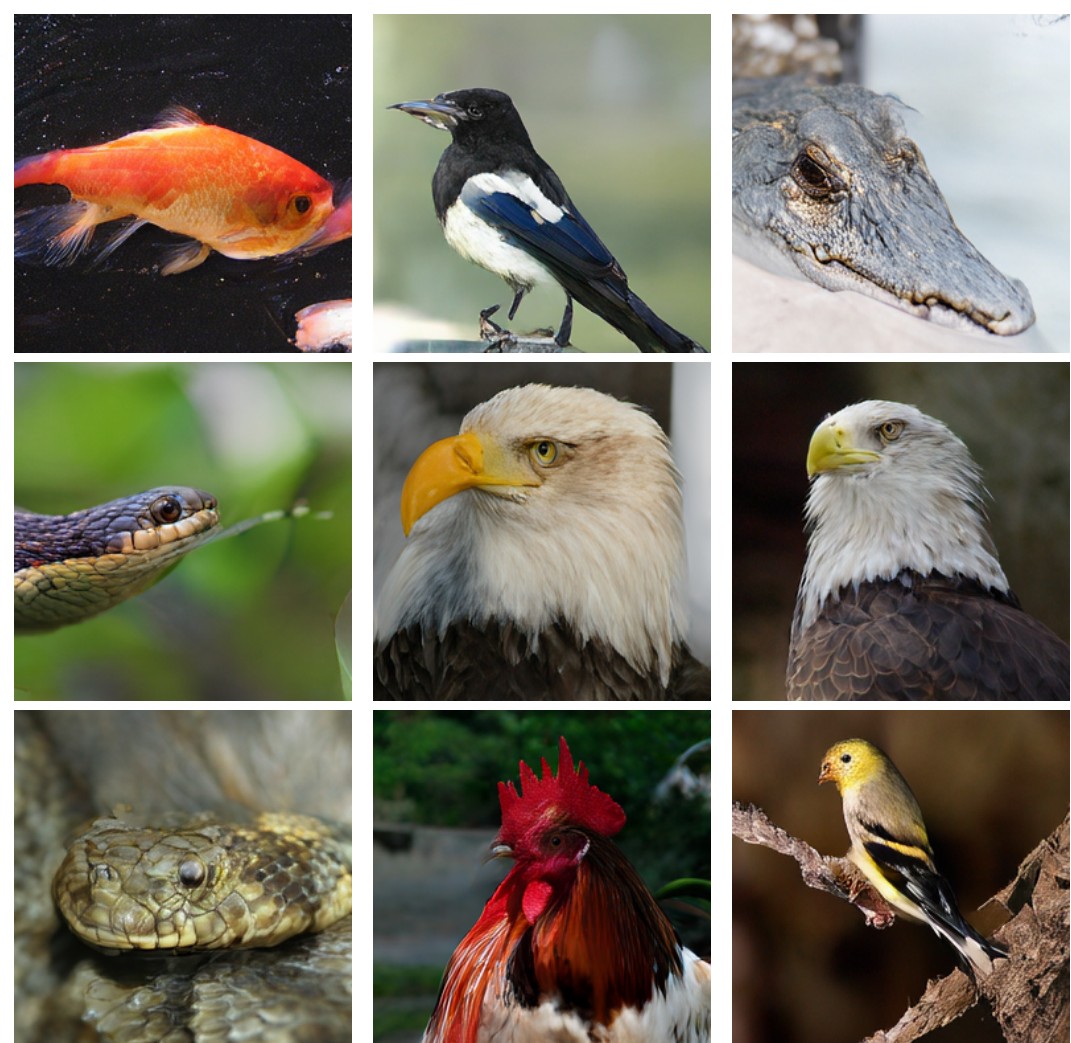

Figure 5: Additional qualitative samples generated by our model.

