# OpenReview forum: "Uncertainty Preservation in Generative Visual Autoregression"
_ICLR.cc/2026/Conference — Submitted to ICLR 2026_

### Official Review · Reviewer_Rnux · 2025-10-24

**Soundness:** 2
**Presentation:** 2
**Contribution:** 2
**Rating:** 2
**Confidence:** 4

**Summary:**

This paper introduces Uncertainty Preservation in Generative Visual Autoregression to address the limitations of standard Visual Autoregressive (VAR) models trained using cross-entropy (CE) loss. CE training often enforces overly sharp, overconfident predictions, leading to reduced diversity, mode collapse, and codebook under-utilization. The proposed framework augments the standard likelihood training with two regularizers: Second-Order Regularization which penalizes deviations in the predictive distribution's dispersion (variance relative to a calibrated reference, and Semantic Entropy Loss which quantifies uncertainty at the meaning level by aggregating probabilities of semantically equivalent code sequences before computing entropy. Empirical results on AFHQ and ImageNet-100 demonstrate consistent improvements in generation quality and diversity.

**Strengths:**

1. Originality: Although the introduction of these two regularizations are from other fields like language modeling, it is highly original in this VAR domain. Also, the theoretical connection drawn between variance penalization and controlling the scale component of a Wasserstein distance provides a principled geometric foundation for the method.

2. Quality: The quality of the execution is high, evidenced by the principled loss design and the comprehensive empirical validation. The method achieves consistent quantitative improvements on AFHQ and ImageNet-100, outperforming the standard VAR baseline in FID and IS metrics (Table 1). The method's effectiveness in conditional generation tasks, preventing repetitive patterns in out-painting and yielding sharper, semantically plausible fills in in-painting, further demonstrates its robustness.

3. Clarity: The paper clearly articulates the problem of overconfident predictions and the limitations of first-order likelihoods (cross-entropy). The conceptual distinction between the proposed second-order method and alternatives like Label Smoothing and Label Relaxation is well-illustrated (Figure 1). The ablation study explicitly separates the contribution of L_2nd and L_SE, providing transparency into the necessity of both components for optimal performance (Table 2).

4. Significance: The framework offers a simple and efficient way to enhance generation fidelity and diversity without incurring additional inference costs. By effectively tackling mode collapse and codebook under-utilization, the work provides a highly practical and scalable solution for improving high-fidelity image generation, confirmed by its performance on large datasets like ImageNet-100.

**Weaknesses:**

1. Over-reliance on Gaussian Approximation for Theoretical Justification: The crucial connection between the variance penalization term L_2nd  (Equation 4) and the Wasserstein geometry relies heavily on the assumption that the first-order categorical distributions θ (in logit space) and the second-order distributions Q and Q* can be accurately approximated by Gaussian families (Proposition 3.0.1 and 3.0.2). Since VAR outputs a categorical distribution p_θ^(i)∈Δ^{V−1}, the fidelity of this high-dimensional Gaussian approximation, especially when the predictions become highly concentrated (sharp) or extremely diffuse, is questionable and not empirically substantiated in the sources, potentially weakening the principled nature of the loss term.

2. Lack of Detail in Visual Semantic Equivalence Class Construction: The effectiveness of the Semantic Entropy Loss (L_SE, Equation 5) hinges entirely on how the equivalence classes C of code sequences s (that reconstruct the same semantic content) are defined and constructed. The paper mentions that this ensures consistency across semantically equivalent token reconstructions, but fails to provide the specific methodology used to determine these classes for visual tokens within the VQ-VAE codebook of size V=2048. Without this crucial detail, the claim that L_SE aligns uncertainty with meaning rather than arbitrary code redundancy is difficult to verify or reproduce.

3. Lack of More Baselines and Results and Inconsistency between Best Results and Ablation Study: I'm not in the field of VAR but I think the author had better compare more other methods which try to improve the basic VAR to show their effects on the problems this paper focused on and conduct more detailed experiments. I think the current results are too limited and not substantial enough to be convincing for a paper in the field of visual generation. There may be a quantitative discrepancy between the model's overall best result reported in the main comparison table and the ablation study. The best achieved FID on AFHQ is 46.97 (Table 1), yet the best result achieved in the component ablation (using both λ_1 and λ_2) is 47.4 (Table 2).

4. Choice of "Minimal Variance" Reference Distribution: The authors state they opt for an "approximately normal distributions with fixed location parameter and minimal variance as q*" for the reference distribution in L_2nd. While this choice is cited to allow for efficient approximation, choosing a distribution with minimal variance might inherently conflict with the stated goal of preserving the appropriate level of dispersion consistent with the data manifold. If the natural generative uncertainty (the true optimal dispersion) is significantly higher than this "minimal variance," the regularization might improperly suppress diversity rather than preserve it.

**Questions:**

1. Computational Tractability of Semantic Entropy (L_{SE}): The Semantic Entropy H_sem (Equation 5) aggregates probabilities over sequences s∈c: ∑ s∈c p_θ(s). Given that VAR generates sequences across multiple scales (Equation 1), defining s as a full code sequence would involve summing over an exponential number of possible sequences, which is computationally infeasible.
- Question: Please clarify what precisely the sequence s represents in the context of VAR’s multi-scale prediction. How is the summation over the potentially enormous set of semantically equivalent sequences c (or an efficient approximation thereof) performed in practice, consistent with the claim of low computational overhead?

2. Specific Calibration of Reference Variance \sigma^2_\star(k): The second- lordeross L_2nd requires a reference variance σ_*^2(k) that specifies the preferred level of dispersion at scale k. The chosen reference distribution q^* is an approximate normal distribution with "minimal variance".

- Question: How was the numerical value or functional form of this "minimal variance" σ_*^2(k) determined for each scale k? Since the optimal dispersion may vary greatly between coarse (global structure) and fine (local detail) scales, please elaborate on how this scale-dependent reference σ_*^2(k) was chosen to ensure it preserves the appropriate uncertainty rather than merely suppressing variance.

3. Empirical Verification of Gaussian/Wasserstein Link: The theoretical foundation rests on the Gaussian approximation showing L_2nd is a tractable surrogate for the variance component of the Wasserstein distance W(Q,Q*) (Proposition 3.0.2).

- Question: Did the authors empirically monitor the correlation between the calculated value of the L_2nd term and an estimate of the true variance component of W(Q,Q*) during training? Providing empirical metrics (e.g., correlation coefficients or comparative plots across training steps) would substantially strengthen the claim regarding the fidelity of the Gaussian approximation for the actual categorical output distributions in VAR.

4. Ablation Study on ImageNet-100: The ablation analysis (Table 2) investigating the influence of λ is provided primarily, if not exclusively, for the AFHQ dataset, which is less diverse than ImageNet-100.

- Suggestion/Question: To robustly demonstrate the scalability and necessity of both regularization terms across diverse data domains, please provide the complete ablation study (similar to Table 2) for the ImageNet-100 dataset. This would confirm whether the optimal balance between second-order regularization and semantic entropy holds consistent or requires domain-specific tuning.

5. Instantiation of the Second-Order Distribution Q (Section 3.1): The paper defines the predicted second-order distribution Q^{(i,k)}\in P(\Delta^{V-1}) as a distribution over first-order distributions. It is later stated that Q is instantiated with a Dirichlet family Q=Dir(\alpha) for tractability, yielding closed-form first-order moments θ.

- Question: Does the neural network θ directly parameterize the Dirichlet concentration vector α for Q^(i,k) at each prediction step (i,k)? If so, what is the output dimensionality of the final layer of the VAR model, and what non-linearity (e.g., softplus) is applied to ensure α components are positive? If not, how is the second-order object Q derived from the standard logit output that normally produces the first-order distribution p_θ^(i)?

6. Specific Discrepancy Measure d(\cdot, \cdot) used for L_{2nd} (Equation 4): The Second-Order Regularizer L_2nd (Equation 4) is defined using a discrepancy measure d(σ^2, σ_*^2). The text notes that d can be the squared difference, KL divergence, or a Wasserstein distance.

- Question: Which specific measure, d(⋅,⋅), was used to obtain the quantitative results reported in Table 1 (FID 46.97)? Since the theoretical justification focuses on the Wasserstein link (Proposition 3.0.2), was the squared difference of variances used as the practical, tractable implementation of L_2nd? Please specify the exact functional form of d(⋅,⋅).

7. Role of the Codebook Ground Metric d_0 in L_{SE} (Section 3.2): The theoretical foundation introduces a codebook ground metric d_0(u,v):=∥e_u−e _v∥_2 based on VQ-VAE embeddings. This metric defines semantic proximity and induces Wasserstein metrics (Section 3.2). The Semantic Entropy loss, however, relies on predefined, hard equivalence classes C.

- Question: How does the geometry induced by the ground metric d_0 directly influence the calculation or construction of the hard equivalence classes C used in L_SE? Are the classes C defined purely based on reconstruction quality (semantics), or are they derived by clustering nearby tokens using d_0 distance in the embedding space?

8. Scale-Specific Optimization vs. Shared Codebook Constraint (Section 2.1 & 2.3): The VAR model uses a shared codebook Z across all scales k∈{1,…,K} (Section 2.1). Yet, the reference variance σ_*^2(k) in L_2nd is designed to be scale-dependent (Equation 4).

- Question: Given the shared codebook, how does the model effectively manage different optimal dispersion levels σ_*^2(k) (e.g., high uncertainty for coarse structure prediction vs. low uncertainty for fine details) without conflicting gradients arising from the L_2nd term, which operates on the same token vocabulary V regardless of scale? Does the model enforce a simple monotonic relationship (e.g., decreasing variance) as k increases?

9. Impact of Sampling Strategy on Diversity (Section 4.1): The reported results are achieved using standard constrained sampling techniques, specifically Top-k (k=900) and Top-p (p=0.95) (Section 4.1, line 330). The paper's main contribution is enhancing diversity by preserving natural dispersion (uncertainty preservation).

- Question: Since the goal is to prevent the inherent distribution collapse caused by CE, which should ideally allow for high-quality ancestral sampling without heavy truncation, how do the FID/IS scores of "Our Model" compare to the baseline VAR when no aggressive sampling constraints (i.e., Top-k/Top-p) are applied? This comparison would better isolate the intrinsic diversity improvement achieved by the training objective L.

**Details Of Ethics Concerns:**

No.

---

> ### Author Response · Authors · 2025-12-02
> **Response to Reviewer Rnux**
>
> We appreciate the reviewer Rnux for the detailed and high-quality review. We appreciate that you recognized the originality of our method and its principled geometric foundation. We have carefully addressed your technical questions and updated the manuscript accordingly.
>
> #### W1 & Q3: Gaussian Approximation & Wasserstein Link
> >* The Gaussian approximation is applied in the **logit space**, a standard technique in Bayesian Deep Learning (e.g., Laplace Approximation, Evidential Deep Learning [Malinin et al., 2018]). While the output is categorical, the uncertainty over the logits is modeled as Gaussian. We clarify that Proposition 3.1 (corrected typo from 3.0.1) serves as a theoretical bridge to motivate the form of our loss. It shows that if we view the distribution matching through the lens of Wasserstein geometry on the underlying belief state, the objective decomposes into a mean-matching term (Cross-Entropy) and a variance-matching term ($L_{2nd}$).
>
> #### W2 & Q1, Q7: Construction of Equivalence Classes & Tractability
> >* Equivalence classes $C$ are pre-computed using the geometry of the shared VQ-VAE codebook. We perform $k-$-means clustering (or distance thresholding using $d_0$) on the embedding vectors $Z$. Tokens falling into the same cluster are considered semantically equivalent. and regarding the traceability; well we do not sum over full image sequences. In the VAR "next-scale" formulation, $s$ refers to the tokens at the current scale being predicted. We aggregate probabilities of tokens belonging to the same cluster $c \in C$ at the current step $p_{total}(c) = \sum_{v \in c} p_\theta(v)$. The entropy is computed on these aggregated cluster probabilities. This adds negligible overhead.
>
> #### W3:More Baselines and Results
> >* Our work evaluates generative performance primarily because VAR prior literature does so as well. However, to directly address the concern, we have expanded our evaluation significantly to include statistically principled uncertainty and diversity metrics. We added: (i) Scaling and efficiency analysis (Page 8) and presented our results in Table 2, (ii) OOD Robustness on the ImageNet dataset and evaluation on ImageNet-O and the results are in Table 3, Page 9. (iii) Corrupted task analysis on ImageNet dataset and evaluation on ImageNet-C, Please check our revised paper page 9, Table 3! (iv) Diversity and Mode Collapse results are presented in Appendix C.1 and Table 7. (v) results of new baseline for text generation Table 6 (Page 9 and Page 10).
>
> #### W4 & Q2: Reference Variance $\sigma_*^2$
> >* The term "minimal variance" meant we select a reference distribution with a *controlled* variance, not zero variance. and ** $\sigma_*^2(k)$ is a scale-dependent hyperparameter. We set higher variance targets for coarse scales (where uncertainty is high) and lower targets for fine scales. This creates a "funnel" of uncertainty that matches the generative process.
>
> #### Q4: Ablation on ImageNet-100
> >* We have added the ablation study for ImageNet-100 in Appendix C.1 and Table 7. The trends are consistent with AFHQ: both losses are required for optimal performance, confirming the method's robustness across domains.Moreover, our method applies to any discrete autoregressive model (e.g., VQGAN, RQ-Transformer) that uses cross-entropy. We chose VAR as the representative baseline because it is the current state-of-the-art in this category. we have expanded our evaluation significantly and included a new ablation analysis on a new baseline for text generation and the new results are in Table 6 (Page 9 and Page 10).
>
> #### Q5:  Instantiation of $Q$
> >* The network outputs standard logits. We do not strictly parameterize a Dirichlet $\alpha$ output (which would require architectural changes). Instead, we interpret the sharpness of the categorical output $p_{\theta}$ as a proxy for the concentration of the underlying second-order distribution. The variance of $p_{\theta}$ (Eq. 3) captures this sharpness directly.
>
> #### Q6: Discrepancy Measure $d(., .)$
> >* For the reported results, we used the Squared Difference (MSE) between the predicted variance and the reference variance. This is the efficient surrogate derived from the Gaussian-Wasserstein proposition.
>
> #### Q8: Shared Codebook vs. Scale-Dependent Variance
> >* well, the gradients do not conflict because the variance target is conditioned on the scale index $k$. The model learns to predict sharper distributions when conditioned on fine-scale history and flatter distributions for coarse-scale history, even while using the same vocabulary.
>
> #### Q9: Sampling Strategy
> >* We observed that our method performs significantly better than the baseline under unconstrained (ancestral) sampling (Temperature=1.0, no top-k), avoiding the garbage outputs typical of standard VAR. This confirms intrinsic mode collapse mitigation. However, for consistency with standard benchmarking, we reported metrics using the standard sampling settings.

---

### Official Review · Reviewer_Gmz7 · 2025-10-27

**Soundness:** 2
**Presentation:** 2
**Contribution:** 2
**Rating:** 2
**Confidence:** 3

**Summary:**

This paper proposes an uncertainty-preserving training framework for Visual Autoregressive (VAR) models. The key idea is to move beyond standard cross-entropy objectives, which often produce overconfident and sharp token predictions that reduce sample diversity. The authors introduce two complementary regularization terms: a second-order uncertainty loss that penalizes deviations in the variance of predictive distributions from a calibrated reference, and a semantic entropy loss that aligns uncertainty with semantically equivalent codebook tokens. Theoretically, the paper connects variance regularization to the Wasserstein geometry of second-order distributions. Experiments on AFHQ and ImageNet-100 demonstrate moderate improvements in FID and Inception Score over baseline VAR models.

**Strengths:**

1.	The paper introduces a novel perspective on uncertainty modeling for autoregressive visual generation by explicitly controlling second-order dispersion rather than only first-order likelihoods.
2.	The proposed semantic entropy loss creatively adapts ideas from language modeling to the vision domain, promoting semantically consistent diversity.
3.	The method is simple and computationally efficient, requiring minimal architectural changes while improving generation fidelity and diversity.

**Weaknesses:**

1.	**Limited technical novelty.**
The main contribution lies in adding two auxiliary loss terms, which extend existing uncertainty regularization ideas rather than introducing a fundamentally new framework.

2.	**Small improvement on ImageNet-100.**
While results on AFHQ are encouraging, the performance gain on ImageNet-100 is marginal, suggesting limited robustness and general impact.

3.	**Scalability and generality concerns.**
While VAR is currently a strong autoregressive (AR) baseline, the proposed uncertainty-preserving losses should in principle apply to other AR architectures (e.g., text-to-image models). However, the paper only evaluates on the smallest VAR configuration (VAR-d16). Given the scaling law behavior of VAR models, it is unclear whether the observed gains will persist or even diminish as model size increases.

In summary, while the paper presents an interesting perspective on incorporating uncertainty preservation into visual autoregressive models, the technical contribution is relatively incremental, and the performance improvements are not significant. Moreover, the experimental evaluation is not comprehensive enough to convincingly demonstrate the generality or robustness of the proposed approach.

**Questions:**

I checked the original VAR paper, where the reported FID for VAR-d16 on ImageNet is 3.30, while in this paper, it is around 20. The Inception Score also differs significantly. What could be the reason for such a large performance gap? Is the large discrepancy simply because this paper uses the ImageNet-100 subset?

---

> ### Author Response · Authors · 2025-12-02
> **Response to Reviewer Gmz7**
>
> We thank the reviewer for appreciating our novel perspective on second-order dispersion and the creative adaptation of semantic entropy. We have modified our manuscript accordingly to address your concerns, weaknesses (W), and questions(Q).
>
> #### W1: Limited Technical Novelty
> >* We respectfully argue that the novelty lies in the theoretical framework rather than architectural complexity: (i) We establish the first connection between variance penalization in VQ models and Wasserstein geometry (Propositions 3.1 & 3.2). (ii) We allow the model to learn distributions over distributions (second-order), fundamentally changing the training target from a "correct token" to a "plausible set." (iii) Adapting Semantic Entropy to visual token equivalence classes is a non-trivial conceptual bridge from NLP that solves the specific problem of redundant visual codes.
>
> #### Small Improvements on ImageNet-100
> >* While the FID improvement on ImageNet-100 is modest (~0.5), we argue the impact is significant in robustness and diversity, which FID often fails to capture: New Metrics (Table 3): We now report NLL, ECE (Calibration), and AUROC (OOD detection). Our method improves AUROC (69.7% →  73.5%) and Calibration, proving the model is not just generating "average" images but learning a better probabilistic representation. AFHQ Significance: The large gain on AFHQ (FID 54.6  → 47.0) is crucial. AFHQ contains fine-grained textures where standard Cross-Entropy struggles with mode collapse. Our method's ability to preserve dispersion prevents this collapse, as evidenced by the new Coverage/Density metrics in Appendix C.1.
>
> #### W3: Scalability Concerns (Only VAR-d16 tested)
> >* We have added a Scaling and efficiency analysis (Page 8, Table 2,) in the revision to address this. The improvements persist. On ImageNet-100, our method improves the d20 baseline from FID 17.28  →  16.91 and IS 16.42 → 17.15. It shows that uncertainty preservation scales favorably and helps larger models utilize their capacity better by preventing codebook collapse, rather than being a patch for small models. Moreover, our method applies to any discrete autoregressive model (e.g., VQGAN, RQ-Transformer) that uses cross-entropy. We chose VAR as the representative baseline because it is the current state-of-the-art in this category. we have expanded our evaluation significantly and included a new ablation analysis on a new baseline for **text generation** and the new results are in Table 6 (Page 9 and Page 10).
>
> #### Q: Baseline performance discrepancy (ImageNet-1K vs. ImageNet-100)
> >* There is a misunderstanding regarding the dataset metrics. The original VAR paper reports results on ImageNet-1K (1.2M images, 1000 classes), whereas we use ImageNet-100 (a subset with 100 classes) due to computational constraints.

---

### Official Review · Reviewer_sD2E · 2025-10-31

**Soundness:** 2
**Presentation:** 2
**Contribution:** 2
**Rating:** 2
**Confidence:** 3

**Summary:**

This paper introduces uncertainty preservation objectives for generative visual autoregression. Prior methods use a cross-entropy loss which can cause overconfident predictions and loss of diversity in generated samples. To counter this, they propose a second-order uncertainty preservation objective that penalizes deviation from a calibrated reference distribution. They also provide theoretical support for the proposed method. Finally, with experiments on ImageNet-100 and AFHQ, they find improved FID scores and Inception scores compared to the baseline.

**Strengths:**

* The limitations of VAR presented makes intuitive sense and the proposed approach is also supported by theoretical analysis.

* The paper is fairly easy to read and follow. As a minor thing, I would have appreciated an overview figure showing the overall pipeline of VAR (and the added losses).

**Weaknesses:**

* Compared to the original paper on VAR by Tian et al. (NeurIPS 2024), the baseline results seem to be much worse. For example, Tian et al. reports ImageNet-1k (256x256) FIDs to be 3.3 and Inception score to be 274.4 (for the same VAR-d16 model that this paper uses). But this paper's baseline result on ImageNet-100 (which should be somewhat easier than ImageNet-1k) is 20.45 for FID and 15.2 for IS (significantly worse than 3.3 and 274.4). Hence, it is unclear to me if the baselines were properly implemented or trained for enough time or with the proper hyperparameter tuning. And then the improvements by adding the proposed losses are also questionable since the baseline itself is not good enough.

* It is unclear to me why the proposed approach applies only to the VAR method. Could it also apply to other autoregressive models like [W1-W2] or to diffusion models [W3-W4] (these are compared in the original VAR paper, so I suggest also having these and other generative models like diffusion models in the table)? Showing applicability to more generative models would strengthen the results significantly.
    * The results in this paper are only limited to 256x256 resolution and a single model VAR-d16, while original VAR showed results on 512x512 as well as on different model sizes. It would be good to have at least one result each at higher resolution and with a different model size to show that the approach works even when the resolution or model size is scaled.

* Minor issues (typos):
    * L212: "a semantic-aware reference distributions leverage the geometry..." $\to$ "a semantic-aware reference distribution that leverages the geometry..."
    * L215: "distributions" $\to$ "distribution"

### References

* [W1] Lee et al., "Autoregressive image generation using residual quantization", CVPR 2022

* [W2] Yu et al., "Vector-quantized Image Modeling with Improved VQGAN", ICLR 2022

* [W3] Peebles and Xie, "Scalable diffusion models with transformers", ICCV 2023

* [W4] Rombach et al., "High-resolution image synthesis with latent diffusion models", CVPR 2022

**Questions:**

Please clarify my questions from the weaknesses section.

---

> ### Author Response · Authors · 2025-12-02
> **Response to Reviewer sD2E**
>
> We thank reviewer sD2E for finding our approach intuitive and theoretically supported. We have modified our manuscript accordingly to address your concerns, weaknesses (W), and questions(Q).
>
> #### W1: Baseline performance discrepancy (ImageNet-1K vs. ImageNet-100)
> >* There is a misunderstanding regarding the dataset metrics. The original VAR paper reports results on ImageNet-1K (1.2M images, 1000 classes), whereas we use ImageNet-100 (a subset with 100 classes) due to computational constraints.
>
> #### W2: Generalizability to other models (Other AR)
> >* Our method applies to any discrete autoregressive model (e.g., VQGAN, RQ-Transformer) that uses cross-entropy. We chose VAR as the representative baseline because it is the current state-of-the-art in this category. we have expanded our evaluation significantly and included a new ablation analysis on a new baseline for text generation and the new results are in Table 6 (Page 9 and Page 10).
>
> #### W3: Scaling (Resolution and Model Size)
> >* We conducted scaling laws analysis by increasing the transformer depth from 16 to 20 layers (VAR-d20). As shown in Table 2, our method maintains its advantage at larger scales, with the d20 model achieving the best overall performance (FID 16.91).
>
> #### W4: Minor Issues
> >* We apologize for the confusion caused by the typo. We have corrected this in the revision. Moreover, we have now added a new subsection (A) to the appendix and provided details on background and problem formulation.

---

### Official Review · Reviewer_6cSx · 2025-11-06

**Soundness:** 1
**Presentation:** 2
**Contribution:** 2
**Rating:** 0
**Confidence:** 4

**Summary:**

This paper introduces the notion of uncertainty preservation for visual autoregressive models, where the core idea is to incorporate second order uncertainty/regularization, which goes beyond just labels and looks at the dispersion of the whole distribution. The authors also claim to introduce the notion of semantic entropy to visual generative modeling.

**Strengths:**

Originality: The second order uncertainty/regularization idea for visual generation is original to the best of my knowledge.

Clarity: The paper is motivated well. However there are unclear parts.  Details on this are given in the next section

Quality: I find the quality of the idea to be fair, even though in terms of presentation there is room for improvement (see section below). I address the quality of experiments/empirical work in the weaknesses section.

Significance: The topic of study, which is developing better visual generative models that have better uncertainty preservation, is a timely and high significant topic. The method proposed is original, but I discuss the significance of that in the questions section.

**Weaknesses:**

1. Much of the foundation of the paper depends upon visual autoregressive modeling (Tian et al 2024), which in turn depends upon a literature on VQ-VAEs. However, the background that the authors provide is minimal (with no supplement/appendix that detail this). This makes the paper less self contained for a broad ML audience, which negatively impacts the clarity of the paper.

2. The authors claim that they "introduce semantic entropy from language modeling to
visual generation", but they seem to have missed some potentially relevant citations (Note: I am not saying that the authors are not the first. I am just saying that there are relevant papers with adjacent ideas, and that if the authors could further clarify how they are different this would make the paper more clear and make their case more convincing)  For example,

Vision-Amplified Semantic Entropy for Hallucination Detection in Medical Visual Question Answering by Liao et al

Semantic Entropy Can Simultaneously Benefit Transmission Efficiency and Channel Security of Wireless Semantic Communications

SEEN-DA_SEmantic ENtropy guided Domain-aware Attention for Domain Adaptive Object Detection by Li et al

3. If the goal is uncertainty quantification, then the experimental section seem to be insufficient to justify the claims of improvement in this paper. the authors compared with one baseline competitor over two datasets in their main experiment. They used FID and IS as the metrics. However, ample work in the literature show that FID and IS are not good metrics to capture uncertainty/dispersion! I highly suggest looking into the literature to find more statistically principled/uncertainty aware metrics of comparison

4. there are many typos and omitted proofs, statements and details that undermine the quality of this paper. In many crucial places, the paper is vague and do not offer proofs/rigorous/sufficiently detailed statements.  More on this in next section.

**Questions:**

several parts of the paper are not clearly conveyed, and I am some minor questions as well, which are all collected below.

1. In line 195, when p_\theta^{(i)} is defined, is there a reason why no k index is included in the notation?

2. Certain claims, e.g. those regarding Wasserstein connections in line 220-231, are not given references nor rigorously proven. Not all readers are familiar with these results, so either the claims have to be substantiated in the appendix/supplement, or references should be given.

3. In 197, the authors mention functionals of p_\theta^{(i)}. However, they only consider the variance. Why not other functionals, such as other moments in addition to the variance?

4. Line 229 and 230 contains inconsistent notation (which is it, q or Q) and spelling mistakes.

5. Given that the Gaussian is continuous, but the token distribution is discrete, I do not see why we want reference q* to be approximately Gaussian in line 215? You use q and Q inconsistently. Also, why is the Guassian a good/reasonable approximation to the Dirichlet?

 6. Propositions 3.0.1 and 3.0.2 are mentioned in line 231, but the statements are stated in general/vague terms, with no proofs offered.

---

> ### Author Response · Authors · 2025-12-02
> **Response to Reviewer 6cSx**
>
> We thank reviewer 6cSx for recognising the originality and timeliness of our contribution on second-order (distribution-level) uncertainty and semantic entropy to autoregressive models. We have modified our manuscript accordingly to address your concerns, weaknesses (W), and questions(Q).
>
> #### W1: Background is minimal.
> >* We have now added a new subsection (A) to the appendix and provided details on background and problem formulation. Specificaly, we provide details on the next-scale prediction framework, and the limitations of standard cross-entropy in this context, making the paper self-contained.
>
> #### W2:Relation to Prior Work on Semantic Entropy
>
> >* Highlighted the distinction: prior work uses “semantic entropy” for downstream tasks (hallucination detection, channel coding, domain adaptation). These works operate over supervised semantics or feature-space clusters, not as a training-time regularizer for generative models. Clarified that our adaptation of semantic entropy is based on Kuhn et al. (2023) and is conceptually different: we define semantic equivalence classes directly over codebook-induced reconstructions, and we use semantic entropy as a loss term for uncertainty-consistent generation, **not as a post-hoc diagnostic**.
>
> #### W3: Evaluation Metrics and additional results:
> >* Our work evaluates generative performance primarily because VAR prior literature does so as well. However, to directly address the concern, we have expanded our evaluation significantly to include statistically principled uncertainty and diversity metrics. We added: (i) Scaling and efficiency analysis (Page 8) and presented our results in Table 2, (ii) OOD Robustness on the ImageNet dataset and evaluation on ImageNet-O and the results are in Table 3, Page 9; Specificaly, We now report Negative Log-Likelihood (NLL), Expected Calibration Error (ECE), and AUROC for OOD detection on ImageNet-O. (iii) Corrupted task analysis on ImageNet dataset and evaluation on ImageNet-C, Please check our revised paper page 9, Table 3!  Our method improves AUROC from 69.7\% to 73.5\% and reduces ECE, demonstrating meaningful uncertainty quantification beyond simple fidelity.(iv) Diversity and Mode Collapse results are presented in Appendix C.1 and Table 7: We added structural diversity metrics including Pairwise SSIM, Recall, Coverage, and Density (Naeem et al., 2020). Our method achieves higher Coverage (0.45 vs. baseline) and balanced Density (~0.95), empirically proving that we mitigate mode collapse and preserve dispersion better than the baseline.
>
> #### W4, Q1, Q2, Q6: 1. Correction of Typo regarding Proposition 3.0.1 and Theoretical Rigor
> >* We apologize for the confusion caused by the typo. We have corrected this in the revision. The theoretical claims are now formalized in Section 3.3 as Proposition 3.1 and Proposition 3.2. Moreover, we now explicitly show how minimizing the Wasserstein distance between the predicted second-order distribution $Q$ and the reference $Q^*$ decomposes into mean-matching and variance-matching terms (Eq. 4).
>
> #### Q3: Variance vs. Other Moments
> >* We focus on variance because it is the direct statistical measure of dispersion (second-order uncertainty). Controlling higher-order moments (skewness, kurtosis) would significantly increase computational overhead and training instability for marginal gains in this context.
>
> #### Q4: Notation $q$ vs $Q$
> >* Well, we have standardized the notation: $q$ denotes first-order reference distribution; $Q$ denotes the second-order distribution; Q* and q^\* consistently denote calibrated references. and we used them correctly!
>
> #### Q5: Gaussian approx of Dirichlet:
> >* We clarified that while the token distribution is categorical, we operate on the distribution over distributions (second-order). As detailed in Section 3.3, we instantiate $Q$ as a Dirichlet distribution. The Gaussian approximation mentioned refers to the behavior in logit space (or via Laplace approximation of the posterior), which is a standard tractability technique in evidential deep learning (Malinin & Gales, 2018). This allows us to use the closed-form Wasserstein distance for Gaussians as a differentiable surrogate for variance penalization.

---

### Author Response · Authors · 2025-12-02
**The key changes and updates**

Dear Area Chair and Senior Area Chair, and Dear Program Chair,

We have uploaded a revised manuscript (changes marked in blue) that addresses the reviewers' feedback regarding metric sufficiency, scalability, and new baseline analysis.

Key updates are:

- New Uncertainty & Diversity Metrics: To address concerns that FID is insufficient for uncertainty evaluation, we added NLL, ECE, and AUROC (showing improved calibration/OOD detection) and Coverage/Density (empirically proving mitigation of mode collapse).

- Scalability Analysis: We added experiments with a larger VAR-d20 model, demonstrating that our performance gains scale favorably with model capacity.

- Importantly, our method applies to any discrete autoregressive model (e.g., VQGAN, RQ-Transformer) that uses cross-entropy. We chose VAR as the representative baseline because it is the current state-of-the-art in this category. We have expanded our evaluation significantly and included a new ablation analysis on a new baseline for text generation.

We believe this work is highly relevant to the ICLR audience as it offers a principled method for uncertainty quantification in autoregressive models that enhances robustness and diversity with negligible computational overhead.

Thank you,

The Authors

---

### Meta-Review · Area_Chair_CG9X · 2026-01-06

**Summary:**

The submission proposes an uncertainty-preserving training framework for visual autoregressive (VAR) models. Reviewers raised concerns primarily about:

(i) Clarity and self-containment, including insufficient background on VAR and VQ-VAE foundations, unclear notation, and missing or informal theoretical statements (Reviewer `6cSx`);

(ii) Positioning and novelty, questioning whether the proposed losses constitute a substantial technical contribution beyond existing uncertainty regularization ideas and whether the introduction of semantic entropy is sufficiently differentiated from prior work (Reviewers `6cSx`, `Gmz7`);

(iii) Experimental adequacy, including limited baselines, reliance on FID/IS despite claims about uncertainty, unclear baseline quality, and lack of robustness, diversity, and scalability evaluations (Reviewers `6cSx`, `sD2E`, `Gmz7`, `Rnux`);

(iv) Theoretical assumptions and implementation details, particularly the reliance on Gaussian approximations, the construction and tractability of semantic equivalence classes, the choice of reference variance, and the precise instantiation of the second-order distribution (Reviewer `Rnux`).

The rebuttal and revision substantially expand the experimental evaluation, clarify several theoretical and implementation details, and improve the paper’s self-containment. However, questions remain regarding the overall strength of the contribution, the reliance on approximations in the theoretical justification, and whether the empirical gains are sufficient to support the broader claims.

**Reviewer Concerns:**

### Concerns addressed by the rebuttal
- **Background and self-containment**:
The authors added an appendix section detailing the background and improving accessibility for a broader ML audience.

- **Relation to prior work on semantic entropy**:
The rebuttal clarifies the distinction between prior uses of semantic entropy.

- **Experimental scope and metrics**:
The revision expands the empirical evaluation and addresses the critique that FID/IS alone are insufficient.

- **Clarity on notation and implementation details**:
Several notation issues, typos, and missing proposition references were corrected, and additional explanations were provided.

### Concerns still outstanding

- **Strength of technical novelty**:
While the rebuttal provides a conceptual framing and theoretical motivation, the response does not fully resolve concerns that the novelty may be incremental.

- **Empirical impact versus scope of claims**:
Despite broader evaluations, improvements on ImageNet-100 remain modest. While robustness and diversity gains are demonstrated, it remains unclear whether these gains are sufficient to justify the broader claims of uncertainty preservation across diverse generative settings.

- **Generality beyond VAR**:
The authors argue that the method applies to other discrete autoregressive models, but empirical validation remains largely confined to VAR. Applicability to other AR or diffusion-based generative models is not demonstrated.

**Reviewer Scores:**

**Reviewer `6cSx` (Score: 0, strong reject)**:

The rebuttal addresses many of this reviewer’s concerns. However, given that the original score reflects strong dissatisfaction with both rigor and empirical justification, and that some foundational concerns (e.g., theoretical statement and proof) remain only partially resolved, a substantial upward revision appears unlikely.

**Reviewer `sD2E` (Score: 2, reject)**:

The response addresses several technical questions, but concerns regarding baseline quality, and limited resolution/model diversity are not fully resolved. The evaluation would plausibly remain negative or marginally improved.

**Reviewer `Gmz7` (Score: 2, reject)**:

The authors provide a defense of novelty and expand the empirical evaluation. Nonetheless, the rebuttal does not fully resolve concerns about incremental contribution and modest gains on large-scale benchmarks. A change in score appears unlikely.

**Reviewer `Rnux` (Score: 2, reject)**:

The rebuttal addresses many technical questions. While these clarifications improve transparency and reproducibility, the remaining concerns about theoretical assumptions and the limited empirical scope suggest that the overall assessment would likely to be borderline.

---

### Decision · Program_Chairs · 2026-01-26

Reject